# Flash-DD: An Ultra Parameter-Efficient Approach to Dataset Distillation

## Abstract

Dataset distillation (DD) aims to create a smaller dataset that encapsulates the essential knowledge of a larger dataset, thereby reducing storage demands and accelerating downstream training. For large-scale dataset distillation, state-of-the-art methods achieve satisfactory performance by using soft labels generated by well-trained teacher models during downstream training. However, it will cause some issues: (1) a substantial amount of additional storage is required to retain the teacher models, often significantly exceeding the storage needed for the synthetic images; (2) generating labels through these teacher models slows down the downstream training process, counteracting the efficiency goals of dataset distillation; and (3) downstream training guided by these teacher models, according to our studies, yields suboptimal performance. Focusing on these drawbacks, in this paper, we propose plug-and-play parameter-efficient label generation techniques for dataset distillation, which maximizes the benefits of limited model parameters and can be generalized to different DD methods, datasets, and settings. Specifically, we propose a DD-oriented model parameter reduction method that automatically determines the optimal capacity of teacher models and eliminates redundant parameters for dataset distillation tasks. Furthermore, for additional parameter space, we turn to model ensemble strategies and propose guidelines to optimize the utilization efficiency of the additional space. Compared to the state-of-the-art methods, Flash-DD requires only **0.03%** of the additional storage and significantly accelerates downstream label generation by **843.81×** while maintaining comparable performance. Alternatively, with a mere **1.8%** storage budget, it can boost accuracy by up to **13.4%** over previous leading methods. Our code will be available.

## 1 Introduction

Dataset distillation (DD) Wang et al. (2018) aims to generate a significantly smaller synthetic dataset that retains the representative knowledge of the original large-scale dataset, maintaining downstream tasks performance while alleviating the storage and transmission burden, and accelerating the process of downstream training. After a period of rapid development in DD field, dataset distillation methods have achieved lossless performance Guo et al. (2023) on low-resolution, small-scale datasets (*e.g.*, CIFAR-10 AND CIFAR-100). However, for high-resolution, large-scale datasets like ImageNet-1K, increased complexity and computational demands make distillation more challenging, leading to significant performance degradation at high compression rates. To mitigate this, state-of-the-art large-scale dataset distillation methods Yin et al. (2024); Yu et al. (2025); Shao et al. (2024); Sun et al. (2024) employ well-defined data augmentation strategies and assign soft labels during downstream training process to enhance data diversity and assure the informative, accurate knowledge for downstream tasks.

However, it will cause some issues. Firstly, a substantial amount of additional storage is required to retain the teacher model(s), which often significantly exceeding the storage needed for the synthetic images and contradicts the storage efficiency benefits provided by dataset distillation. Additionally, it will introduce extensive GPU time and computational overhead for incorporating soft label generation process during downstream training. To make more accurate knowledge transfer and convey more information to the downstream tasks, each augmented image will be passed through the teacher model(s) at every iteration to generate the corresponding soft label, which considerably slows down

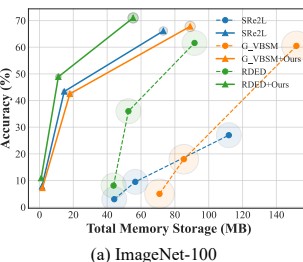 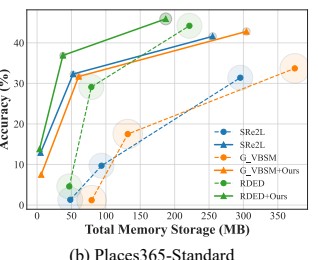 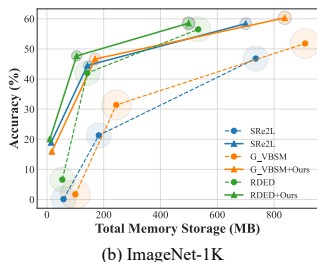

(a) ImageNet-100        (b) Places365-Standard        (b) ImageNet-1K

Figure 1: Comparison of downstream performance and required storage (both synthetic images and teacher model) for state-of-the-art methods: SRe$^2$L, G_VBSM, RDED, and those methods with Ours. Points from left to right correspond to IPC 1, 10, and 50. Circle size denotes the GFLOPs of teacher model used in downstream training.

the training process. It also counteracts the efficiency goals of dataset distillation. Lastly, as illustrated in Fig. 1, downstream training guided by the well-trained teacher model(s) yields suboptimal performance, further highlighting the limitations of current soft label generation techniques.

In this paper, we propose plug-and-play, memory-efficient soft label generation techniques for dataset distillation. Our method also establishes a paradigm, offers optimal solutions under different storage conditions and is adaptable to various dataset distillation methods, datasets, and settings. Specifically, we propose a DD-oriented model parameter reduction method that automatically determines the optimal capacity of teacher models to downstream tasks. Further, to optimize the spatial efficiency of the additional parameter space, we systematically restructure the parameter space to extend and optimize the initially small effective portion across the entire capacity, and also the eliminate potential negative effects raised by the redundant and irrelevant parameters. Here, we turn to model ensemble strategies to satisfy the requirements of the different downstream training stages and provide guidelines to obtain feasible and diverse knowledge for downstream tasks. Extensive experiments validate the efficacy of our proposed method. For example, with just 0.03% of the original extra storage, we maintain the performance while speed up the label generation process 843.81×. Also, under the constant limited storage, our proposed method outperforms the previous state-of-the-art method up to 13.4%.

To sum up, our contributions are listed as follows:

- We propose plug-and-play parameter-efficient soft label generation techniques for dataset distillation to fully leverage the benefits of the limited model parameters. We also establish a paradigm for achieving optimal solutions for different storage conditions, which can be applied across different DD methods, datasets, and settings.

- We propose a DD-oriented model parameter reduction method that automatically determines the optimal capacity of teacher models for downstream training and removes the task-irrelevant parameters to accelerate the training process and reduce the storage space.

- We systematically re-organize the additional parameter space to optimize the spatial efficiency. We propose guidelines for model ensemble strategies to adapt to the different downstream training stages while providing feasible and diverse knowledge.

- We conduct extensive experiments to validate the effectiveness of our method. Results demonstrate that our method only requires 0.03% extra storage to get the same performance of the previous state-of-the-art methods while accelerating the soft label generation 843.81×. With 1.8% extra storage budget, our method surpasses the previous state-of-the-art methods up to 13.4%.

## 2 RELATED WORKS

Dataset distillation, first proposed by Wang *et al.* Wang et al. (2018), distills the large-scale and complex original dataset into a simpler and much smaller synthetic one without compromising its performance. After a period of development, the field of dataset distillation has undergone explo-

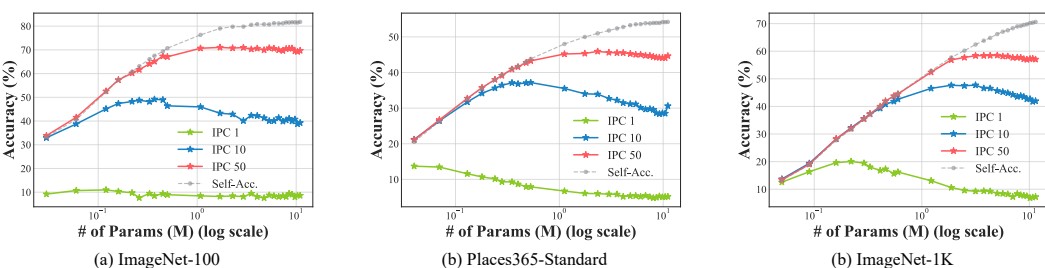

(a) ImageNet-100      (b) Places365-Standard      (b) ImageNet-1K

Figure 2: Illustration of the relationships between the downstream performance, the size of the teacher, and the performance of the teacher. The experiments are conducted on three datasets ImageNet-100, Places365-Standard, and ImageNet-1K, with IPC 1, 10, and 50. For better demonstration, the # of Params (MB) is set to a log scale.

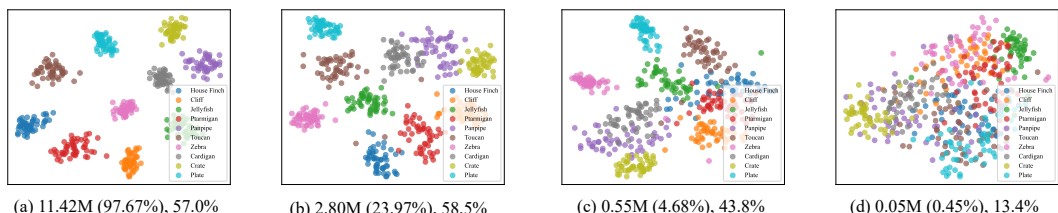

(a) 11.42M (97.67%), 57.0%    (b) 2.80M (23.97%), 58.5%    (c) 0.55M (4.68%), 43.8%    (d) 0.05M (0.45%), 13.4%

Figure 3: The t-SNE analysis of ResNet-18 under different storage budgets on ImageNet-1K with IPC 50. Each subfigure title shows model size, its ratio to the original model (in parentheses), and the corresponding downstream accuracy. The analysis is performed on the models pre-softmax logits.

ration improvements in many aspects. For instance, in terms of optimization objectives, some methods Wang et al. (2018); Nguyen et al. (2020; 2021); Zhou et al. (2022); Loo et al. (2022; 2023); Deng & Russakovsky (2022) optimize the distilled datasets according to the model performance across the original dataset and the synthetic one as defined by dataset distillation. Some methods Zhao et al. (2020); Zhao & Bilen (2021); Cazenavette et al. (2022); Lee et al. (2022); Du et al. (2023); Cui et al. (2023); Lee & Chung (2024); Shin et al. (2023) optimize the distilled datasets by the training trajectories or loss curve of the model training process on those two datasets, while some Wang et al. (2022); Zhao et al. (2023); Zhao & Bilen (2023); Sajedi et al. (2023); Zhang et al. (2024) pursue optimization by aligning the distributions or attention maps of the datasets. Moreover, numerous efforts have contributed to progress in directions such as dataset parameterization Kim et al. (2022); Cazenavette et al. (2023); Liu et al. (2022a); Su et al. (2024), label distillation Bohdal et al. (2020); Sucholutsky & Schonlau (2021).

However, dataset distillation often suffers from performance drops on downstream tasks when applied to large-scale datasets. Recent state-of-the-art methods Yin et al. (2024); Shao et al. (2024); Yu et al. (2025); Sun et al. (2024); Shao et al. (2025) address this by using pre-trained teacher model(s) to generate soft labels either online or in advance for each epoch during downstream training. This enriches and strengthens the synthetic dataset by leveraging teacher knowledge, but also increases storage requirements and prolongs downstream training, contrary to the original goals of dataset distillation. To address these issues, our proposed method offers training-efficient and memory-flexible solutions. Specifically, we introduce a DD-oriented model parameter reduction technique that automatically determines the optimal teacher capacity for downstream tasks, reducing storage needs and accelerating training. Additionally, we provide guidelines for adaptive model ensemble strategies in soft label generation, further improving parameter efficiency and downstream performance.

## 3 METHOD

### 3.1 PRELIMINARY

For large-scale dataset distillation, state-of-the-art large-scale dataset distillation methods Yin et al. (2024); Shao et al. (2024); Yu et al. (2025); Sun et al. (2024) adopt augmentation strategies and

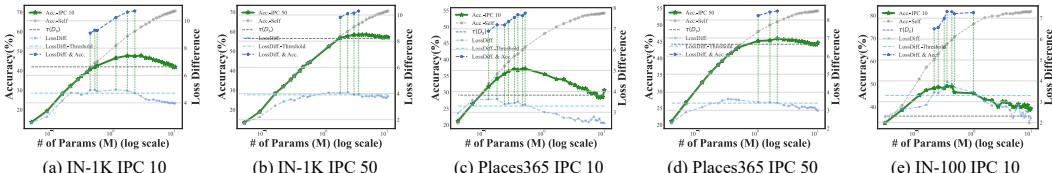

| (a) IN-1K IPC 10 | (b) IN-1K IPC 50 | (c) Places365 IPC 10 | (d) Places365 IPC 50 | (e) IN-100 IPC 10 |

Figure 4: The validation experiments for the criteria to determine the optimal capacity for teacher models. The experiments are conducted under the settings of ImageNet-1K and Places365-Standard with IPC 10 and 50 and ImageNet-100 with IPC 10. Here, for ImageNet-1K and Places365-Standard, we calculate the loss reduction in the early 50 epochs for downstream training, and $\gamma$ here is 0.1. For ImageNet-100, the process is in the early 20 epochs and $\gamma$ is 0.05.

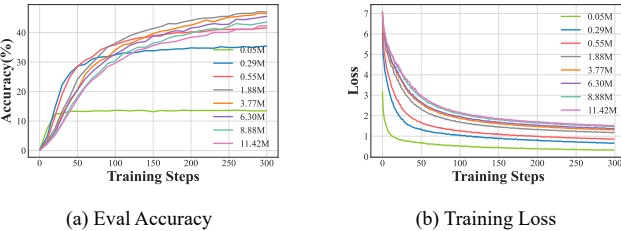

| (a) Eval Accuracy | (b) Training Loss |

Figure 5: The downstream evaluation loss (a) and training loss (b). The evaluation is under ImageNet-1K with IPC 10.

employ teacher models to generate soft labels for each augmented image during downstream training to compensate for the loss of knowledge:

$$
\begin{aligned}
\theta_S^{(t)} &= \theta_S^{(t-1)} - \alpha \nabla \mathcal{L}((X_s^{(t)\prime}, Y_s^{(t)\prime}); \theta_S^{(t-1)}), \\
X_s^{(t)\prime} &= \mathcal{A}ug(X_s, t; \phi), Y_s^{(t)\prime} = f_{\theta_T \sim \Theta_T}(X_s^{(t)\prime}; \theta_T).
\end{aligned}
\tag{1}
$$

Here, $\theta_S$ is the downstream model, $\alpha$ is the learning rate, and $\mathcal{L}(\cdot; \theta_S)$ is the loss function for downstream tasks. During the downstream training process, the synthetic images $X_s$ will be augmented by $\mathcal{A}ug$ with the parameter $\phi$ to generate the training samples $X_s^{(t)\prime}$ for epoch $t$, and $|X_s^{(t)\prime}| = |X_s|$. The corresponding labels $Y_s^{(t)\prime}$ are generated by the teacher model(s) $\Theta_T$. Although soft labels greatly improve downstream performance, they also increase storage requirements for teacher model(s) and introduce downstream training latency. Our method seeks an optimal balance among training efficiency, storage, and performance.

## 3.2 DD-ORIENTED MODEL PARAMETER REDUCTION

Current state-of-the-art large-scale dataset distillation methods leverage teacher models to enhance downstream task performance. However, this introduces significant extra storage demands and slows down downstream training. To address these issues, we propose a DD-oriented model parameter reduction method that optimizes teacher model capacity, enabling efficient and effective soft label generation for better downstream generalization. The framework of our method is shown in Algorithm 1.

We start with a full-sized $\theta_T$ teacher model pretrained on the entire original dataset. Our method consists of two steps. First, to obtain a lightweight teacher with equivalent accuracy, we iteratively prune the full model using $l_1$-norm importance scores until performance begins to degrade.

$$
\hat{I_{g,k}} = I_{g,k} / \beta_g,
\tag{2}
$$

where $I_{g,k} = \sum_{w \in g} ||w[k]||_1$, the $l_1$-norm importance score of the k-th prunable dimension of the parameter $w \in g$. $\beta_g$ is the average $l_1$-norm importance score of the top-N most important parameter $w \in g$. This step will quickly remove the redundant parameters while maintain the performance of both the teacher model and the downstream tasks.

---

**Algorithm 1** DD-Oriented Model Parameter Reduction

---

**Input**: Synthetic dataset $D_S$, full-sized teacher $\theta_{T_0}$, $\tau(D_S)$
**Output**: Capacity matched $\theta_T^*$

1: $i = 0$
2: **while** $\mathcal{Acc}(\theta_{T_i}) \geq \mathcal{Acc}(\theta_{T_0})$ **do**
3:    $\theta_{T_{i+1}} = Param\_reduce(\theta_{T_i})$
4:    $i = i + 1$
5: **end while**
6: $\Theta_{\mathcal{T}} = \{\}$
7: **while** $\mathcal{Acc}(\theta_{T_i}) > \tau(D_S)$ **do**
8:    $\theta_{T_{i+1}} = Param\_reduce(\theta_{T_i})$
9:    Compute the loss degradation during the early training stage $\mathcal{Dis}(\theta_{T_i})$.
10:    **if** $\frac{\partial^2 \mathcal{Dis}(\theta_{T_i})}{\partial |\theta_{T_i}|^2} = 0$ **then**
11:       $\zeta = \mathcal{Dis}(\theta_{T_i})$
12:    **end if**
13:    **if** $\mathcal{Dis}(\theta_{T_i}) \geq \zeta$ **then**
14:       $\Theta_T \cup \theta_{T_i}$ // Only consider the loss degradation more than the threshold $\zeta$.
15:    **end if**
16:    $i = i + 1$
17: **end while**
18: $\theta_T^* = \arg\max_{\theta_T} \mathcal{Dis}(\theta_T) + \gamma \mathcal{Acc}(\theta_T), \theta_T \in \Theta_T$
19: **return** $\theta_T^*$

---

The next step is to determine the optimal amount of teacher knowledge that students can effectively utilize. To achieve this, we exclude challenging or non-essential knowledge based on the learnable space and knowledge capacity of the synthetic datasets, enhancing their generalization across different IPCs and teacher models. We conduct experiments with teachers of varying knowledge capacities across different synthetic dataset sizes, as shown in Fig. 2. The results show that downstream performance initially increases with more teacher knowledge but subsequently declines, indicating that the redundant teacher knowledge contributes little to improving performance; instead, only a small portion takes effect. Additionally, we perform t-SNE analysis on the outputs of models with varying pruning rates, as shown in Fig. 3. The teacher model that yields optimal downstream performance provides clear class boundaries and well-preserved inter-class relationships, thus offering richer information. Furthermore, when teacher model performance falls below a specific threshold, the model obtained through downstream transfer aligns with the teacher performance. Here, we refer to this threshold as $\tau(Ds)$, $Ds$ refers to the distilled dataset.

**Proposition 1.** $\tau(Ds)$ *is the accuracy of the downstream student models trained with the guidance of the original pre-trained full-size teacher model on distilled dataset $D_S$.*

*Proof.* Here, for simplicity, we assume the case of linear model. Denote the original pre-trained full-size teacher model as $T_0$, with parameter $\theta_{T_0}$. The teacher model with parameter reduction as $T_1$, with parameter $\theta_{T_1}$. $S_i$ with parameter $\theta_{S_i}$ is student models trained under the guidance of $T_i$, $n$ is the size of the distilled dataset $D_S$.

$$\theta_S^* = \arg\min_{\theta_S} \frac{1}{n} \sum_{i=1}^{n} ||f(x_i, \theta_T) - \theta_S^T \cdot x_i||^2 + \lambda ||\theta_S||^2,$$

$$\theta_S^* = (X^T X + \lambda I)^{-1} X^T \cdot f(X, \theta_T), \quad (3)$$

where $X \in \mathbb{R}^{n \times d}$. When $n << d$, $r(X^T X) \leq n$, $\theta_S^* \approx \frac{1}{\lambda} X^T \cdot f(X, \theta_T)$. Denote the test error of student model on the test set:

$$\varepsilon(S) = \mathbb{E}_{(x,y) \sim D_{test}}[(y - (\theta_S^*)^T x)^2] \approx \mathbb{E}[(y - f(x, \theta_T))^2] + \mathbb{E}[(f(x, \theta_T) - (\theta_S^*)^T x)^2]. \quad (4)$$

The error is divided into two parts, the teacher test error and the guidance error. For the distillation error,

$$\varepsilon_{dis} = \mathbb{E}_{(x,y) \sim D_{test}}[(f(x, \theta_T) - (\theta_S^*)^T x)^2] \geq \mathbb{E}[f(x, \theta_T)^2] - \frac{||X^T f(X, \theta_T)||^2}{\lambda^2} \mathbb{E}[x^T x]. \quad (5)$$

As in $n \ll d$, $\varepsilon_{dis} \geq \varepsilon(T) - \mathcal{O}(\frac{n^2}{\lambda^2 d}) \approx \varepsilon(T)$, the performance of the student model depends on the quality of the teacher, and the student cannot reduce the error. Thus, $\varepsilon(S_0) = \varepsilon(T_1) \rightarrow \varepsilon(S_1) = \varepsilon(S_0)$. $\qquad\square$

This rule indicates the lowest bound to obtain the optimal capacity of the teacher model for downstream tasks, and provides insights to find the lowest memory costs to obtain comparable state-of-the-art performance.

By analyzing the downstream results and training dynamics (Fig. 5), we identify key criteria for automatically determining teacher capacity to maximize downstream performance. First, teachers with accuracy below $\tau(Ds)$ should be excluded, as theoretically analyzed before. Also, for downstream training,

$$
\begin{aligned}
\mathcal{L} &= \min \mathbb{E}_{(x,y)\sim\mathcal{D}_s}[||f(x,\theta_T) - f(x,\theta_S)||_2^2] + \eta\mathbb{E}_{(x,y)\sim\mathcal{D}_s}[||f(x,\theta_S) - y||_2^2] \\
&= \min(1+\eta)\mathbb{E}[||f(x,\theta_T) - f(x,\theta_S)||_2^2] + \eta\mathbb{E}||f(x,\theta_T) - y||_2^2 + 2\eta\Delta\cdot\epsilon,
\end{aligned}
\tag{6}
$$

where $\Delta = f(x,\theta_T) - f(x,\theta_S)$ and $\epsilon = f(x,\theta_T) - y$. The performance of $\theta_S$ is closely related to the performance of $\theta_T$. For teachers with a low knowledge capacity, students can fit the outputs more easily with a small amount data $D_S$. However, when the teacher model exhibits poor performance and deviates substantially from the ground truth distribution, the resulting improvement is limited. In contrast, well-aligned teachers yield greater improvements. As shown in Fig. 5, student difficulty in fitting teacher outputs can be identified early in downstream training.

$$
\begin{aligned}
\mathcal{D}is(\theta_T) &= \mathbb{E}_{\theta_S^{(0)}\sim\mathcal{P}_\theta}[\mathcal{L}(D_S; \theta_S^{(t)}, \theta_T) - \mathcal{L}(D_S; \theta_S^{(0)}, \theta_T)], \\
\theta_T^* &= \arg\max_{\theta_T} \mathcal{D}is(\theta_T) + \gamma\mathcal{A}cc(\theta_T).
\end{aligned}
\tag{7}
$$

Here, $t$ represents epoch in the early training stage, $\mathcal{L}(\cdot)$ is the loss function for the downstream tasks. $\mathcal{D}is(\cdot)$ calculates the loss reduction during the early stage. $\mathcal{D}is(\theta_T) > \zeta$, $\zeta$ is the threshold for the loss reduction, to filter out the teachers with either excessively high (difficult to fit) or excessively low knowledge capacity. Analyzed from experimental results, the threshold can be obtained by:

$$
\zeta = \{\mathcal{D}is(\theta_T)|\frac{\partial^2\mathcal{D}is(\theta_T)}{\partial|\theta_T|^2} = 0\}.
\tag{8}
$$

We also conduct the validation experiments to demonstrate the effectiveness of the criteria, as shown in Fig. 4.

### 3.3 GUIDELINES FOR ENSEMBLE LABEL GENERATION

Previously, we demonstrate the effectiveness of aligning teacher and synthetic dataset knowledge capacity for downstream tasks. In this section, we focus on ensemble label generation and propose guidelines for better transferability. This approach avoids redundant or ineffective knowledge which may degrade performance and provides more informative guidance for downstream tasks, enabling maximal use of available space. Specifically, we restructure the pre-defined space $M$ into several sub-modules and incorporate additional perspectives from other teachers:

$$
f(x; \overline{\Theta}_T^{(t)}) = \sum_{\theta_{T_i}\in\Theta_T} \mu_i^{(t)} f(x; \theta_{T_i}).
\tag{9}
$$

Here, $M = \sum_{\theta_{T_i}\in\Theta_T} \mathcal{C}(\theta_{T_i})$, $\mathcal{C}(\cdot)$ measures the size of the model. $\overline{\Theta}_T^{(t)}$ represents the ensembled teachers for epoch $t$. $\mu_i^{(t)}$ is the weight for model $\theta_{T_i}$ for epoch $t$, and it could be updated during the downstream training optionally.

To achieve effective subspace division, as mentioned in section 3.2, teacher models should have appropriate knowledge capacity for better downstream transfer. Using an ensemble of teacher models introduces diverse, informative perspectives, but significant discrepancies in their guidance (e.g., conflicting interpretations of the same instance) should be minimized. As shown in Fig. 3, models with insufficient capacity may provide inconsistent guidance. Thus, when capacity of each teacher

| Methods | | ImageNet-100 | | | Places365-Standard | | | ImageNet-1K | | |
|---|---|---|---|---|---|---|---|---|---|---|
| | | 1 | 10 | 50 | 1 | 10 | 50 | 1 | 10 | 50 |
| SRe²L | Acc. | $3.0 \pm 0.3$ | $9.5 \pm 0.4$ | $27.0 \pm 0.4$ | $1.3 \pm 0.2^*$ | $9.7 \pm 0.1^*$ | $31.4 \pm 0.1^*$ | $0.1 \pm 0.1$ | $21.3 \pm 0.6$ | $46.8 \pm 0.2$ |
| | Extra Mem. | 42.8MB | 42.8MB | 42.8MB | 43.3MB | 43.3MB | 43.3MB | 44.7MB | 44.7MB | 44.7MB |
| w/ Ours | Acc. | $7.8 \pm 0.6$ ↑4.8 | $43.4 \pm 0.1$ ↑33.9 | $66.1 \pm 0.1$ ↑39.1 | $12.9 \pm 0.2$ ↑11.6 | $32.3 \pm 0.3$ ↑22.6 | $41.6 \pm 0.1$ ↑10.2 | $18.9 \pm 0.7$ ↑18.8 | $44.5 \pm 0.1$ ↑23.2 | $58.4 \pm 0.1$ ↑11.6 |
| | Extra Mem. | 0.1MB 0.3% | 0.8MB 2.0% | 4.1MB 9.6% | 0.1MB 0.3% | 2.0MB 4.6% | 3.1MB 7.1% | 1.1MB 2.5% | 4.7MB 10.5% | 9.5MB 21.3% |
| G_VBSM | Acc. | $5.0 \pm 0.1^*$ | $18.0 \pm 0.4^*$ | $60.5 \pm 0.1^*$ | $1.2 \pm 0.1^*$ | $17.5 \pm 0.1^*$ | $33.7 \pm 0.6^*$ | $1.7 \pm 0.1^*$ | $31.4 \pm 0.5$ | $51.8 \pm 0.4$ |
| | Extra Mem. | 69.3MB | 69.3MB | 69.3MB | 73.4MB | 73.4MB | 73.4MB | 84.1MB | 84.1MB | 84.1MB |
| w/ Ours | Acc. | $7.2 \pm 0.6$ ↑2.2 | $42.5 \pm 0.2$ ↑24.5 | $67.8 \pm 0.6$ ↑7.3 | $7.5 \pm 0.3$ ↑6.3 | $31.7 \pm 0.2$ ↑14.2 | $42.8 \pm 0.7$ ↑9.1 | $15.9 \pm 0.1$ ↑14.2 | $46.6 \pm 0.2$ ↑15.2 | $60.3 \pm 0.1$ ↑8.5 |
| | Extra Mem. | 0.1MB 0.2% | 1.9MB 2.7% | 6.5MB 9.4% | 0.1MB 0.2% | 2.0MB 2.7% | 3.1MB 4.2% | 1.1MB 1.3% | 9.5MB 11.3% | 12.1MB 14.4% |
| RDED | Acc. | $8.1 \pm 0.3$ | $36.0 \pm 0.3$ | $61.6 \pm 0.1$ | $4.6 \pm 0.2^*$ | $29.1 \pm 0.3^*$ | $44.2 \pm 0.2^*$ | $6.6 \pm 0.2$ | $42.0 \pm 0.1$ | $56.5 \pm 0.1$ |
| | Extra Mem. | 42.8MB | 42.8MB | 42.8MB | 43.3MB | 43.3MB | 43.3MB | 44.7MB | 44.7MB | 44.7MB |
| w/ Ours | Acc. | $10.9 \pm 0.3$ ↑2.8 | $48.9 \pm 0.3$ ↑12.9 | $71.0 \pm 0.2$ ↑9.4 | $13.8 \pm 0.1$ ↑9.2 | $36.9 \pm 0.2$ ↑7.7 | $45.9 \pm 0.1$ ↑1.7 | $20.0 \pm 0.1$ ↑13.4 | $47.5 \pm 0.2$ ↑5.5 | $58.5 \pm 0.1$ ↑2.0 |
| | Extra Mem. | 0.04MB 0.1% | 1.4MB 3.3% | 6.5MB 15.2% | 0.1MB 0.3% | 2.0MB 4.6% | 8.9MB 20.6% | 0.8MB 1.8% | 7.2MB 16.1% | 10.7MB 23.9% |

Table 1: Performance and efficiency comparison with baseline large-scale dataset distillation methods across three datasets ImageNet-100, Places365-Standard, and ImageNet-1K. Here, the extra memory represents for the storage part excluding the synthetic images. All synthetic datasets are generated by official code without further update. ResNet-18 is adopted for both image generation and evaluation processes. * represents that the results are from our reproduction.

| Datasets | | #Params ↓ | Extra Mem. ↓ | GFLOPs ↓ | Speed Up ↑ | Acc. (Teacher) | Acc. (Downstream) |
|---|---|---|---|---|---|---|---|
| **ImageNet-100** | 1 | <0.006M (0.05%) | <0.02MB (0.05%) | <0.003 (0.14%) | >729.31× | <16.26 | $9.5 \pm 0.3$ |
| | 10 | 0.04M (0.37%) | 0.16MB (0.37%) | 0.01 (0.60%) | 166.15× | 36.22 | $35.8 \pm 0.1$ |
| | 50 | 0.26M (2.28%) | 0.98MB (2.28%) | 0.05 (2.96%) | 33.73× | 63.10 | $61.6 \pm 0.3$ |
| **Places365-Standard** | 1 | 0.003M (0.03%) | 0.01MB (0.03%) | 0.002 (0.12%) | 843.81× | 4.09 | $4.1 \pm 0.1$ |
| | 10 | 0.08M (0.74%) | 0.32MB (0.74%) | 0.02 (1.11%) | 90.12× | 29.00 | $29.2 \pm 0.1$ |
| | 50 | 0.75M (6.59%) | 2.86MB (6.59%) | 0.14 (7.55%) | 13.32× | 45.66 | $44.3 \pm 0.1$ |
| **ImageNet-1K** | 1 | 0.03M (0.22%) | 0.10MB (0.22%) | 0.004 (0.16%) | 609.94× | 6.49 | $6.6 \pm 0.1$ |
| | 10 | 0.55M (4.68%) | 2.09MB (4.68%) | 0.09 (4.84%) | 20.66× | 43.85 | $41.8 \pm 0.1$ |
| | 50 | 1.88M (16.12%) | 7.19MB (16.12%) | 0.30 (16.24%) | 6.16× | 57.73 | $57.0 \pm 0.1$ |

Table 2: Results on the least extra memory costs to achieve previous state-of-the-art method RDED performance. We conduct experiments on three datasets ImageNet-100, Places365-Standard, and ImageNet-1K for IPC 1, 10, and 50. Here, we report the number of parameters, required extra memory costs, GFLOPs, and accuracy of the teacher model. Also, we evaluate the acceleration rate for the soft label generation part (the extra required time for downstream training), and the performance for downstream tasks.

is comparable, and with feasible knowledge, leveraging additional different perspectives can offer benefits for downstream tasks. The criteria can be formulated as follows:

$$\max_{\theta_{T_i} \in \Theta_T} \mathcal{C}(\theta_{T_i}) - \min_{\theta_{T_i} \in \Theta_T} \mathcal{C}(\theta_{T_i}) < \epsilon_1, \quad \sum_{\theta_{T_i}, \theta_{T_j} \in \Theta_T} |\mathcal{C}(\theta_{T_i}) - \mathcal{C}(\theta_{T_j})| < \epsilon_2,$$

$$\forall \theta_{T_i} \in \Theta_T, \ \mathcal{A}cc(\theta_{T_i}) > \tau(D_s), \forall \theta_{T_i} \in \Theta_T, \ |\mathcal{C}(\theta_{T_i}) - \mathcal{C}(\theta_T^*)| < \epsilon_3.$$

(10)

Here, $\epsilon$ refers to the small constant, and $\theta_T^*$ refers to the single teacher which has the optimal downstream performance.

## 4 EXPERIMENT

### 4.1 EXPERIMENT SETTING

#### 4.1.1 DATASET AND NETWORK

We adopt full-sized ImageNet-100, Places365-Standard Zhou et al. (2017), and ImageNet-1K Deng et al. (2009) to validate our method. Here, synthetic datasets are generated by official code without further update. For networks, we adopt ResNet-18 for the synthetic dataset generation and the soft label generation. For baseline comparison, we employ the ResNet-18 as the downstream model for all compared methods. We evaluate the generalization ability of our proposed method on ShuffleNet-

| Datasets | | ShuffleNet-V2 | MobileNet-V2 | EfficientNet-B0 | Swin-V2-Tiny | RegNet-Y-400MF | AlexNet |
|---|---|---|---|---|---|---|---|
| Places365-Standard | RDED | 17.3 ± 0.9* | 20.6 ± 0.5* | 26.1 ± 0.3* | 13.9 ± 0.3* | 26.0 ± 0.3* | 10.3 ± 0.1* |
| | Ours | 27.0 ± 0.5 ↑ 9.7 | 30.6 ± 0.3 ↑ 10.0 | 36.7 ± 0.2 ↑ 10.6 | 20.2 ± 0.2 ↑ 6.3 | 33.5 ± 0.5 ↑ 7.5 | 11.3 ± 0.1 ↑ 1.0 |
| ImageNet-1K | RDED | 23.6 ± 0.5* | 34.4 ± 0.2 | 42.8 ± 0.5 | 17.8 ± 0.1 | 38.5 ± 0.5* | 11.9 ± 0.1* |
| | Ours | 30.7 ± 0.1 ↑ 7.1 | 41.5 ± 0.5 ↑ 7.1 | 47.4 ± 0.1 ↑ 4.6 | 27.5 ± 0.7 ↑ 9.7 | 42.9 ± 0.5 ↑ 4.4 | 14.4 ± 0.1 ↑ 2.5 |

Table 3: Results on cross-architecture generalization evaluation. Here, we conduct experiments under the settings of Places365-Standard and ImageNet-1K with IPC 10. RDED adopts ResNet-18 for both the synthetic image and the label generation parts, while our proposed method adopts ResNet-18 with parameter reduction for label generation.

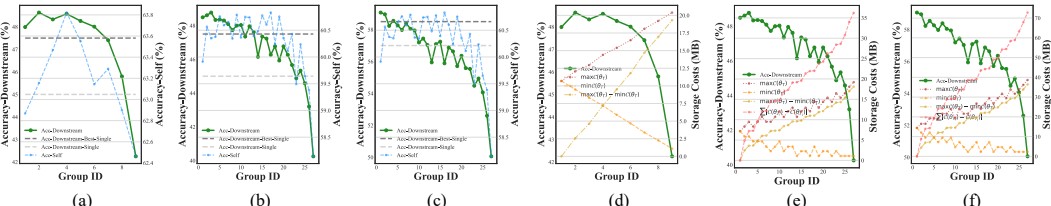

Figure 6: The relationships between the downstream accuracy and the ensemble accuracy (the first three figures) and the parameters of the teachers (the rest ones). (a) (d) are under the settings of ImageNet-1K with IPC 10, the number of teachers is 2, the storage space is 21.65MB; (b) (e) are under the settings of ImageNet-1K with IPC 10, and the number of teachers is 3, the storage space is 21.65MB; (c)(f) are under the settings of ImageNet-1K with IPC 50, the number of teachers is 3, the storage space is 43.66MB.

V2 Ma et al. (2018), MobileNet-V2 Sandler et al. (2018), EfficientNet-B0 Tan & Le (2019), Swin-V2-Tiny Liu et al. (2022b), RegNet-Y-400MF Radosavovic et al. (2020), and AlexNet Krizhevsky et al. (2012).

### 4.1.2 IMPLEMENTATION DETAIL

We evaluate Flash-DD compared with SOTA large-scale DD methods, including SRe²L Yin et al. (2024), G_VBSM Shao et al. (2024), and RDED Sun et al. (2024), reporting both performance and additional storage for baseline comparisons with repeated 3 times. To ensure fairness, soft labels are generated online, and teacher models are treated as extra storage. We report the best performance of our method and its corresponding memory cost. We also present the minimum storage and efficiency needed to achieve previous state-of-the-art results. Here, speed-up denotes the reduction in extra time for soft label generation. For cross-architecture generalization, we use the optimal performance case for comparison. For continual learning, we follow previous works and use the GDumb framework and report the optimal results. For ensemble soft label generation, we conduct experiments on ImageNet-1K IPCs of 10 and 50, under storage constraints of 21.65MB (48.43%) and 43.66MB (97.67%). More experimental details are provided in supplementary.

### 4.2 RESULTS ON BASELINES

The baseline comparisons are divided into two parts. First, we compare our method's optimal performance and extra storage with the baselines in Table 1. For fairness, the teacher model(s) are counted as extra storage for all methods. Second, we report the minimal extra memory needed to achieve previous state-of-the-art performance and evaluate the speed-up in soft label generation. Results are shown in Table 2. For the first part, our method outperforms previous state-of-the-art methods in all settings while greatly reducing resource use. Notably, it brings all baselines to a similar high performance. For example, on ImageNet-1K IPC 50, our method improves the three baselines to 58.4%, 60.3%, and 58.5% with significant reduced extra memory. In the second part, our method achieves state-of-the-art performance with minimal extra storage and greatly accelerates soft label generation. For example, on Places365-Standard IPC 1, it needs just 0.03% of the original extra space and speeds up soft label generation by 843.81×.

| Steps | 1 | 2 | 3 | 4 | 5 |
|---|---|---|---|---|---|
| **RDED** | 22.11 | 21.85 | 25.30 | 26.68 | 29.20 |
| **Ours** | 32.07 | 34.06 | 35.62 | 35.59 | 36.90 |

Table 4: The results on the continual learning tasks. We conduct the experiments under the settings of Places365-Standard with IPC 10.

### 4.3 RESULTS ON CROSS-ARCHITECTURE GENERALIZATION

We evaluate our method on various architectures, including ShuffleNet-V2, MobileNet-V2, EfficientNet-B0, Swin-V2-Tiny, RegNet-Y-400MF, and AlexNet, to demonstrate its generalization ability. Experiments on Places365-Standard and ImageNet-1K with IPC 10 are summarized in Table 3. Our method shows strong generalization across all architectures. Notably, on transformer-based models like Swin-V2-Tiny, it outperforms previous SOTA by 6.3% and 9.7%.

### 4.4 RESULTS ON ENSEMBLE SOFT LABEL GENERATION

For ensemble soft label generation, we conduct experiments on ImageNet-1K with limited storage: 21.65MB (48.43% of the original) using IPC 10 with 2 and 3 teachers, and 43.66MB (97.67%) using IPC 50 with 3 teachers, to demonstrate our method effectiveness. We evaluate different ensemble divisions to identify optimal space utilization and examine relationships between teachers, such as parameter count differences. Results are shown in Fig 6.

As shown in Fig. 6, our method achieves 48.6% with 2 teachers and 48.8% with 3 teachers using 21.65MB storage on ImageNet-1K IPC 10, surpassing the best single teacher (47.5%) by 1.1% and 1.3%, and outperforming the single teacher with 21.65MB (45.0%) by 3.6% and 3.8%. With IPC 50, 43.66MB, and 3 teachers, our method reaches 59.1%, exceeding the best single teacher (58.5%) and the single model with 43.66MB (47.0%). The results also reveal that downstream performance trends the self-performance of the ensembled teachers show a certain level of consistency; divisions with excessively low self-performance should be avoided. Notably, the sum of parameter differences among teachers negatively correlates with downstream performance. For optimal results, teacher models should have similar sizes.

### 4.5 RESULTS ON CONTINUAL LEARNING

Continual learning is an essential application of DD. To demonstrate the effectiveness of our proposed method, we conduct experiments applying our method to the continual learning tasks under the setting of Places365-Standard with IPC 10. Here, we follow the previous work Zhao & Bilen (2023); Yin et al. (2024), adopting the continual learning framework GDumb Prabhu et al. (2020). The results are shown in Table 4. From the results, our proposed methods significantly surpass the previous state-of-the-art method. For each step, our methods show higher performance.

## 5 CONCLUSION

In this paper, we conduct a comprehensive exploration of parameter-efficient soft label generation techniques for dataset distillation to amplify the benefits of limited model parameters for label generation. To be specific, we propose a DD-oriented model parameter reduction method to automatically adapt the model capacity to the purpose of the downstream tasks. Further, we propose guidelines for ensemble soft label generation method to optimize the efficiency of the utilization for the additional parameter space. Experiments demonstrate the efficacy of our method, which achieves comparable state-of-the-art performance with only 0.03% of the original extra storage space while accelerating $843.81\times$ of the label generation process. Also, with 1.8% storage budget, our method surpasses the previous state-of-the-art methods up to 13.4%.

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

## A  APPENDIX

### A.1  DATASET AND NETWORKS

In this paper, we explore the parameter-efficient soft label generation techniques for dataset distillation in large-scale datasets era. Here, to validate the effectiveness of our proposed method, we adopt the widely-used public datasets ImageNet-1K Deng et al. (2009), its subset ImageNet-100, and Places365-Standard Zhou et al. (2017), all of which are full-sized. To obtain the ImageNet-100, we follow the classes ids adopted in previous dataset distillation methods IDC Kim et al. (2022) and RDED Sun et al. (2024). During the process, the images are resized to 224×224.

For downstream tasks training, we generate the synthetic datasets using the official code provided by the baseline methods. Our proposed method is focused on the label generation part, and during downstream training, the synthetic data will not be updated.

We utilize ResNet-18 He et al. (2016) as the base model for parameter reduction and ensemble label generation tasks. For downstream evaluation, ResNet-18 is employed for baseline comparisons and continual learning tasks, with the architecture derived from the official torchvision codebase. For cross-architecture generalization evaluation, ResNet-18 with parameter reduction serves as the teacher model to generate soft labels for downstream tasks. Regarding the baseline methods SRe$^2$L Yin et al. (2024), G_VBSM Shao et al. (2024), RDED Sun et al. (2024), we adhere to the official scripts provided by the authors. For evaluation, we employ a diverse set of models, including ShuffleNet-V2 (X0_5)Ma et al. (2018), MobileNet-V2Sandler et al. (2018), EfficientNet-B0 Tan & Le (2019), Swin-V2-Tiny Liu et al. (2022b), RegNet-Y-400MF Radosavovic et al. (2020), and AlexNet Krizhevsky et al. (2012).

Table 5: The hyper-parameters of pruning and finetuning.

| Hypeparameter | Value |
|---|---|
| Optimizer | SGD |
| Learning Rate | 1e-2 |
| Weight Decay | 1e-4 |
| Batch Size | 64 |
| Finetuning Epochs | 90 |
| Scheduler | StepLR |
| LR Step Size | 30 |
| Pruning Criteria | Magnitude L1 |

Table 6: The hyper-parameters of downstream evaluation and adaptive ensemble label generation.

| Hypeparameter | Value |
|---|---|
| Optimizer | AdamW |
| Learning Rate | 1e-3 |
| Batch Size | 100 |
| Training Epochs | 300 |
| Scheduler | CosineannealingLR |
| Augmentation | CutMix & ShufflePatches |
| Loss Type | MSE + CE |
| CE Weight | 0.025 |
| Learning Rate $\gamma$ (optimal) | 1e-3 |

## A.2 IMPLEMENTATION DETAILS

For dataset distillation-oriented parameter reduction, we utilize the magnitude-based L1 criterion as the basic pruning rule. For the model fine-tuning phase, we employ the SGD optimizer with a learning rate of 1e-2, fine-tuning the model for 90 epochs. Detailed hyper-parameters are provided in Table 5. For determining the best capacity-matched model, we calculate the loss reduction over the first 50 epochs, setting $\alpha$ to 0.1 for ImageNet-1K and Places365-Standard. For the ImageNet-100 dataset, the loss reduction is calculated over the first 20 epochs, with $\alpha$ set to 0.05.

For baseline comparison, we evaluate the performance and memory overhead of our proposed method against state-of-the-art large-scale dataset distillation approaches, including SRe$^2$L Yin et al. (2024), G_VBSM Shao et al. (2024), and RDED Sun et al. (2024). For fairness, we consider teacher models as supplementary storage and assume that soft labels are generated online during the downstream training process. The evaluation is conducted in two stages. First, we present key metrics highlighting the optimal performance of our method, such as the additional memory required. We also analyze the least storage costs and efficiency metrics necessary to achieve comparable performance with prior state-of-the-art methods.

To quantify computational advantages, we define the speed-up metric as the reduction in additional time required for downstream tasks compared to baseline methods. All efficiency experiments are performed on the NVIDIA RTX A5000 GPU to ensure consistency and fairness.

To evaluate cross-architecture generalization, we assess the performance of synthetic datasets and label generator(s) across six diverse architectures. To ensure fairness, we use the efficient generator identified by our method as optimal.

For experiments on continual learning tasks, we employ the GDumb framework Prabhu et al. (2020), consistent with prior works. Comparisons are made under both optimal performance and minimal memory usage scenarios, all within the Places365-Standard IPC 10 setting.

The performance of the ensemble-based soft label generation component is evaluated on ImageNet-1K under IPC 10 and IPC 50, considering pre-defined storage capacities of 21.65MB (48.43%) and 43.66MB (97.67%).

For downstream performance evaluation, we use the generated distilled datasets to train randomly initialized student models, subsequently evaluate them on the corresponding datasets. In line with official experimental setups of comparable methods, all baselines use ResNet-18 for evaluating downstream performance and efficiency. The hyper-parameter settings for downstream evaluation are detailed in Table 6.

## A.3 LIMITATIONS

Our proposed method Flash-DD requires extra time in dataset generation process to obtain the optimal capacity teacher models. In this paper, we propose a DD-oriented parameter reduction method to significantly reduce the extra storage space costs for labels. Also, a light-weight teacher model with optimal capacity is able to accelerate the downstream training while achieving state-of-the-art performance. However, acquiring optimal capacity teacher model involves an iterative accumulation loss calculation process, which introduces extra time overhead to the distilled dataset generation part. Nevertheless, once the optimal capacity teacher model is obtained, this overhead become negligible compared with acceleration achieved in downstream training.

## A.4 BROADER IMPACTS

The field of dataset distillation aims to significantly reduce the required data volume for model training, while maintaining the downstream performance. Thus, for downstream training, it can greatly decreasing storage demands and computational resource consumption. Our proposed method Flash-DD further advances dataset distillation methods from the perspective of label generation, providing further improvements in final performance, storage efficiency, and computational cost for downstream tasks. These improvements not only contribute to the dataset distillation field, but also contribute to more efficient and scalable model training, more efficient use of resources and lowers the environmental impact of deploying advanced AI models.

## A.5 ETHICS STATEMENT

We confirm that this study follows the ICLR Code of Ethics. This research does not involve human participants, personally identifiable information, or sensitive content. All datasets are publicly available and legally distributed. Our contributions aim to improve computational efficiency in dataset distillation for downstream training.

## A.6 REPRODUCIBILITY STATEMENT

We emphasize reproducibility throughout this work. The main content and Appendix provide comprehensive explanations of the proposed method, including training scripts, model architectures, and evaluation settings. All datasets utilized are publicly accessible, and we also provide the pre-processing steps during downstream training for clarity. To further support replication, we include algorithmic pseudocode, detailed parameter settings, and implementation details. Source code and supporting materials are also included to allow verification for our work.

## A.7 THE USE OF LARGE LANGUAGE MODELS

We use large language models only for correcting typographical, grammatical, and spelling mistakes.

