# OpenReview forum: "Flash-DD: An Ultra Parameter-Efficient Approach to Dataset Distillation"
_ICLR.cc/2026/Conference — Submitted to ICLR 2026_

### Official Review · Reviewer_z3eu · 2025-10-25

**Soundness:** 3
**Presentation:** 2
**Contribution:** 3
**Rating:** 6
**Confidence:** 4

**Summary:**

The paper presents FLASH-DD, a plug-and-play and parameter-efficient technique for label generation in dataset distillation. The method aims to improve the teacher model’s storage and effectiveness by reducing redundant parameters and employs an ensemble of models to generate soft labels for training.

**Strengths:**

1. The proposed FLASH-DD requires only 0.03% additional storage and accelerates downstream label generation by 843.81×, while maintaining comparable performance.
2. The proposed method outperforms previous state-of-the-art approaches in all settings and substantially reduces resource usage.

**Weaknesses:**

1. The explanation of Equation (2) on page 4 lacks sufficient detail. It is unclear how the importance score β is calculated, why certain parameters are considered redundant, and what algorithm this step is based on as the theory support.
2. The proof of Proposition 1 on page 5 is insufficiently explained. The notation in Equation (3) is not well introduced, and the underlying theory is missing. It is unclear what Equation (3) represents and what each term denotes.
3. The description of the ensembling strategy in lines 316–317 on page 6 is incomplete. It is unclear whether multiple teacher models are used to generate labels or if submodules from different models are combined to form a new teacher model.

**Questions:**

Does the parameter reduction refer to removing a portion of the parameters from the teacher model? If so, how is the remaining part of the teacher model utilized after this reduction?

---

> ### Author Response · Authors · 2025-11-29
> **Response to Reviewer z3eu**
>
> ### [Weaknesses 1] Further explanation of Equation (2) on page 4.
> We sincerely thank the reviewer for pointing out this issue. In Equation (2), our intention is to use the l1-norm magnitude pruning criterion[1] **as we state in lines 208-215** in the paper, which is one of the most widely used and fundamental pruning methods. While the manuscript mentioned the use of l1-norm, we realize that our explanation around the importance score $\beta$ could have been improved to describe more clearly, and we truly appreciate the reviewer for helping us identify this point.
>
> Specificaly, for l1-norm magnitude structure pruning, the importance of each parameter is assessed by its l1 norm: a larger l1 magnitude indicates higher contribution to the network, whereas a smaller l1 magnitude suggests a higher likelihood of redundancy and potential pruning. In Equation (2), our formulation specifies how the importance of each prunable unit $k$ within a structural group $g$ is computed using this l1-norm based criterion:  $\hat{I_{g,k}}=\frac{I_{g,k}}{\beta_g}$. Here, as we mentioned in the paper, $I_{g,k}=\sum_{w \in g}||w[k]||_1$ denotes the l1-norm of all parameters belonging to unit $k$ inside group $g$. This value directly represents the importance of unit $k$ under the standard magnitude-pruning assumption.
>
> However, the raw importance score $I_{g,k}$ only enables comparison within a group, because different layers and groups naturally have different l1 scales. To make the importance scores comparable across groups and layers, a normalization step is required. In classical l1-norm-based pruning, a common practice is to normalize by the sum average of the top-N largest l1 magnitudes within the group, which provides a stable, scale-invariant estimate of the group's overall importance.
>
> Following this standard practice, in Equation (2) we compute $\beta_g=\frac{1}{N}\sum_{k\in topN(g)}||w[k]||$, and use $\beta_g$ to normalize $I_{g,k}$. This ensures that units from different groups can be compared on a consistent scale while remaining fully aligned with the classical l1-norm magnitude pruning framework.
>
> In addition, we evaluate the performance of our method under different pruning or importance criteria, including l1-norm, l2-norm, and Taylor (gradient-based) pruning. All three approaches produced similar teacher capacities and comparable downstream accuracies across IPC settings. The results for ImageNet-1K with IPC 1, 10, and 50 are shown in the following table.
>
> |**ImageNet-1K**|Taylor|l2-norm|l1 (in paper)|
> |-|-|-|-|
> |IPC 1|19.7 $\pm$ 0.2|19.6 $\pm$ 0.1|20.0 $\pm$ 0.1|
> |Extra Mem.|0.8MB|0.8MB|0.8MB|
> |IPC 10|47.3 $\pm$ 0.2|47.4 $\pm$ 0.1|47.5 $\pm$ 0.2|
> |Extra Mem.|7.2MB|7.2MB|7.2MB|
> |IPC 50|59.2 $\pm$ 0.1|59.1 $\pm$ 0.1|58.5 $\pm$ 0.1|
> |Extra Mem.|10.7MB|10.7MB|10.7MB|
>
>
> These results show that our proposed method is robust with different pruning strategies and maintains stable performance when adopting alternative pruning methods.
>
> [1] Pruning Filters for Efficient ConvNets. Hao Li, Asim Kadav, Igor Durdanovic, Hanan Samet, Hans Peter Graf. International Conference on Learning Representations, 2017.

---

> ### Author Response · Authors · 2025-11-29
> **Response to Reviewer z3eu (part 2)**
>
> ### [Weaknesses 2] Further explanation of Equation (3) on page 5.
> We are truly grateful to the reviewer for highlighting this important issue, which significantly helped us refine and clarify the presentation of our manuscript. Here, for simplicity, we assume the case of linear model for downstream student models with paramter $\theta_S\in\mathbb{R}^{d\times C}$, where $C$ is the number of classes. Specifically, Equation (3) implements the closed-form ridge-regression solution induced by the distillation loss. To enhance readability, we will provide a more explicit introduction of the notation and briefly describe its role before presenting the proposition. Concretely, given a distilled dataset $D_S$, we denote the feature of distilled data as $X=[x_1^T,x_2^T,\dots,x_n^T]^T\in \mathbb{R}^{n\times d}$, $x_i\in \mathbb{R}^d$, $n$ is the number of samples in the distilled dataset, and $d$ is the feature dimension for the distilled data. $f(x_i,\theta_T)\in \mathbb{R}^C$ denotes the output of the teacher model, and $f(X,\theta_T)\in \mathbb{R}^{n\times C}$, and the output for the student model can be expressed as $\theta_S^Tx_i\in\mathbb{R}^{C}$, and $X\theta_S\in\mathbb{R}^{n\times C}$. Here, we start with Equation (3):
>
> $\mathcal{L}(\theta_S)=\frac{1}{n}\sum_{i=1}^n||f(x_i, \theta_T)-\theta_S^Tx_i||^2_2+\lambda||\theta_S||^2_2\rightarrow\frac{1}{n}||f(X,\theta_T)-X\theta_S||^2_2+\lambda||\theta_S||^2_2$,
>
> the $\lambda||\theta_S||^2_2$ is the regulariation term, and $\lambda$ is the regularization coefficient. Then, the final student $\theta^*_S$ will be obtained by the following process.
> First, the gradient of the loss function over $\theta_S$ should be 0:
>
> $\nabla_{\theta_S}\mathcal{L}=\frac{2}{n}X^T(X\theta_S-f(X,\theta_T))+2\lambda\theta_S=0$
>
> $(X^TX+n\lambda I)\theta_S=X^Tf(X,\theta_T)\rightarrow(X^TX+\lambda I)\theta_S=X^Tf(X,\theta_T)$, set $\lambda=n\lambda$.
>
> Thus, the final student is
>
> $\theta_S^*=(X^TX+\lambda I)^{-1}X^Tf(X,\theta_T)$,
>
> as shown in Equation (3).
> When $n \ll d$, we have $\operatorname{rank}(X^\top X) \le n$, so $X^\top X$ is low-rank. If $\lambda$ dominates the nonzero eigenvalues of $X^\top X$, we can approximate
> $(X^\top X + \lambda I)^{-1} \approx \frac{1}{\lambda} I$
> and thus
> $\theta_S^* \approx \frac{1}{\lambda} X^\top f(X,\theta_T)$.
>
>
>
> ### [Weaknesses 3] Further description of the ensembling strategy on page 6.
> We sincerely thank the reviewer for valuable feedback, and we truly appreciate the opportunity to clarify this aspect of our method. To reiterate, our proposed method performs prediction-level ensembling over multiple pruned teacher models $\theta_{T_i}\in\Theta$, as formally defined in **Equation (9)** of the manuscript (**in lines 312-319**). For the predefined storage $M$, the output of the ensembled teachers is:
>
> $f(x;\overline{\Theta}^{(t)})=\sum_{\theta_{T_i}\in\Theta}\mu_i^{(t)} f(x;\theta_{T_i})$
>
> where $M=\sum_{\theta_{T_i}\in\Theta}\mathcal{C}(\theta_{T_i})$, $\mathcal{C}(\cdot)$ is the size of the model. $f(x;\theta)$ is the output of the model with parameters $\theta$ given input $x$. $\overline{\Theta}^{(t)}$ represents the ensembled teachers for epoch $t$. $\mu_i^{(t)}$ is the weight for model $T_i$ with parameters $\theta_{T_i}$ for epoch $t$, which can optionally be updated during downstream training. This formulation indicates that each pruned teacher $T_i$ independently produces a full prediction, and the final output of the ensembled teachers is the weighted average of these outputs.
> As suggested by the guideline in Equation (10), each teacher model is turned to have similar size, so a uniform initialization is a natural choice.
>
> ### [Question] Does the parameter reduction refer to removing a portion of the parameters from the teacher model? If so, how is the remaining part of the teacher model utilized after this reduction?
> We sincerely thank the reviewer for this valuable question and appreciate the opportunity to clarify.
>
> Yes, the parameter reduction in our method refers to structured pruning, where we remove certain structural components of the original full teacher model (e.g., channels, filter groups, or blocks) based on their importance scores. After pruning, the remaining part of each teacher model is fully preserved and continues to function as a valid teacher network with reduced parameters.
>
> Each pruned teacher performs its own forward pass to produce a complete prediction. During downstream training, these predictions are combined through a weighted average, where each teacher contributes according to its weight $\mu_i^{(t)}$. This weighted ensemble output serves as the final supervision signal for downstream training.

---

### Official Review · Reviewer_fTh4 · 2025-10-26

**Soundness:** 2
**Presentation:** 3
**Contribution:** 3
**Rating:** 6
**Confidence:** 4

**Summary:**

The paper proposes Flash-DD, a parameter-efficient dataset distillation framework that reduces teacher model size via automated parameter pruning and uses ensemble-based label generation to optimize knowledge transfer. It achieves up to 843× faster label generation and comparable or superior performance with only 0.03–1.8% of the storage required by prior methods on large-scale datasets like ImageNet-1K.

**Strengths:**

1. The research direction of improving inference efficiency for soft label generation is important, and the authors present an intuitive and effective approach to address this problem.

2. The paper is well structured and clearly written, making it easy to follow.

3. The experimental evaluation is comprehensive and solid, including cross-architecture generalization analysis.

**Weaknesses:**

1. Algorithm 1 lacks clarity in the definition and initialization of the threshold $\zeta$.  The variable $\zeta$ is used in the comparison $Dis(\theta_{T_i}) \ge \zeta$ before being explicitly defined,  which introduces ambiguity in the algorithm's logical flow and practical implementation.

2. The theoretical analysis in Section 3.2, particularly the definition of the threshold $\zeta$ via the second derivative of $Dis(\theta_T)$, lacks mathematical rigor.  Since $Dis(\theta_T)$ is empirically measured rather than analytically defined, the condition $\frac{\partial^2 Dis(\theta_T)}{\partial|\theta_T|^2} = 0$ is largely heuristic.  While identifying an inflection point in teaching effectiveness is intuitively reasonable, the formal expression appears ad hoc and may overstate the theoretical grounding of the method.

3. Code is not provided.

4. (Minor) Figure 1 is somewhat difficult to read; improving its visual clarity would help.

5. (Minor) The legend in most of the figures (Figure 4 and Figure 6) in this paper is so small and hard to read, and the caption doesn't contain full information for understanding the figure itself.

**Questions:**

I like the paper overall and would be happy to raise my score if the authors address the few weaknesses and questions.

1. It is unclear whether the training configurations of the baseline methods (SRe2L, G-VBSM, and RDED) are identical to those used when applying Flash-DD. By “training configurations,” I refer to the hyperparameters and the number of teacher models involved in generating soft labels. For example, SRe2L appears to employ a single ResNet-18 as the teacher model, whereas Flash-DD utilizes multiple teacher models for soft label generation. This raises concerns that the observed performance improvement may partially result from the ensemble effect rather than the proposed parameter-efficient design.

2. A similar concern arises in the comparison with RDED, where RDED uses a single ResNet-18 for label generation, while Flash-DD adopts multiple teacher models. It would be helpful if the authors could include an ablation study or additional analysis to clarify whether the performance gains are primarily due to the ensemble setup or the proposed optimization strategy.

3. The algorithm defines the threshold $\zeta$ using the condition  $\frac{\partial^2 Dis(\theta_T)}{\partial|\theta_T|^2}=0$,  but in practice $Dis(\theta_T)$ is empirically measured rather than analytically defined.  Could the authors elaborate on how $\zeta$ is computed numerically and whether this threshold is robust to noise in the measured $Dis$ values?

4. How is Acc in Algorithm 1 defined? Is it the accuracy of the model on the distilled dataset?

5. How sensitive is the optimal teacher capacity to dataset domain or model architecture? Would a teacher optimized on ImageNet-1K remain near-optimal when transferred to a related dataset such as ImageNet-100?

---

> ### Author Response · Authors · 2025-11-29
> **Response to Reviewer fTh4**
>
> ### [Weaknesses 1 & Weaknesses 2 & Questions 3] Elaboration on how $\zeta$ is computed numerically and whether this threshold is robust to noise in the measured Dis values.
> We sincerely appreciate the reviewer's thoughtful and detailed comments regarding the definition and use of the threshold $\zeta$. For clarity, we provide additional details on how $\zeta$ is defined and computed in our method.
>
> In our framework, $\zeta$ marks the point at which the accumulated loss curve begins to flatten during the teacher parameter reduction process. In Section 3.2, the notation is intended to convey this flattening behavior. We are grateful to the reviewer for kindly pointing this out. For the numerical computation of $\zeta$, $Dis(\theta_{T})$ is evaluated at discrete capacity levels, $\zeta$ can be computed numerically by examining the finite difference ratio: $\frac{\Delta Dis}{\Delta |\theta_T|}$ and identifying the point where the incremental increase becomes small. Here, $\Delta Dis=Dis(\theta_{T_{i+1}})-Dis(\theta_{T_{i}})$, and $\Delta \theta_T=|\theta_{T_{i+1}}|-|\theta_{T_{i}}|$. We select $\zeta$ at the point where this incremental gain becomes sufficiently small, indicating diminishing returns in accumulated loss as the teacher is further reduced.
>
> To further examine robustness, we conduct an additional experiment on ImageNet-100 with IPC 10, in which we expand the candidate capacity range beyond the region indicated by $\zeta$, including several capacities that would normally fall outside this range. As shown in the following table, the teacher selected by Eq. 7 remains unchanged, and the downstream accuracy is minimally affected. This indicates that $\zeta$ serves to narrow the search space, while the overall method remains stable with respect to variations in this threshold.
>
> |Teacher Size (# Param)|$Dis(\theta_T)+\gamma Acc(\theta_T)$|Downstream Acc|
>  |-|-|-|
>  |2.28 $^*$|6.76|42.9 $\pm$ 0.2|
>  |1.70 $^*$|6.83|43.3 $\pm$ 0.2|
>  |1.08|7.28|46.0 $\pm$ 0.1|
>  |0.50|7.22|46.4 $\pm$ 0.1|
>  |0.45|7.30|48.8 $\pm$ 0.2|
>  |0.37|**7.32**|**48.9 $\pm$ 0.3**|
>  |0.33|6.99|48.2 $\pm$ 0.2|
>  |0.26|6.56|48.7 $\pm$ 0.2|
>  |0.22|6.49|48.2 $\pm$ 0.1|
>  |0.16 $^*$|5.72|47.4 $\pm$ 0.1|
>  |0.12 $^*$|5.35|45.1 $\pm$ 0.2|
>
>  *Here, * refers to the capacities that would normally fall outside the range. $\gamma$ is 0.05, the same as we adopt in paper.*
>
>
> We will incorporate these clarifications and the additional results into the next manuscript, and we sincerely thank the reviewer again for helping us strengthen the clarity of this part of the paper.
> ### [Weaknesses 3] Code is not provided.
> We sincerely appreciate the reviewer's suggestion. We ensure that the full implementation will be released after the final decision.
> ### [Minor Weaknesses 4 & Minor Weaknesses 5] Improving the visual clarity in paper.
> We sincerely thank the reviewer for this helpful suggestion. Based on the reviewer's suggestion, we will carefully revise these figures and captions in the next version. We truly appreciate the reviewer's attention to these details, and these revisions will substantially improve the clarity and presentation quality of the paper.

---

> ### Author Response · Authors · 2025-11-29
> **Response to Reviewer fTh4 (part 2)**
>
> ### [Questions 1 & Question 2] Clarification of training configurations for the baseline methods (SRe2L, G-VBSM, and RDED) and our proposed Flash-DD
> We sincerely thank the reviewer for raising this question. We greatly appreciate the opportunity to clarify this point in detail.
>
> First, we would like to emphasize that all results reported in **Tables 1–4 and Figures 1–5** are obtained using **a single pruned teacher**.
> Only Figure 6 (Section 4.4) involves multiple-teacher ensembling results.
> We will revise the manuscript to explicitly highlight this distinction to avoid any misunderstanding in the next version.
>
> Then, we would like to briefly restate the motivation behind our method.
> Flash-DD is designed to fully utilize different available storage budgets in a parameter-efficient manner. Our approach contains two complementary stages:
>
> 1. Single-teacher selection under small budgets.
> As illustrated in Figure 2, when evaluating single pruned teachers of different capacities on downstream tasks, the accuracy typically first increases and then decreases across three datasets and three IPC settings (1/10/50). When the storage budget is very limited (especially when it does not exceed the capacity of the best single teacher), we adopt a single pruned teacher and search for its optimal capacity.
>
> 2. Teacher ensembling under larger budgets.
> As the available storage budget increases, simply enlarging a single teacher does not yield higher accuracy (even may cause performance degradation for complex and difficult knowledge for downstream tasks), and using only the best single-teacher capacity results in substantial unused avaiable space.
> To better utilize the larger budget and improve performance, we introduce in Section 3.3 an ensemble guideline for pruned teachers, which describes how to allocate a given storage budget across multiple pruned teachers to obtain improved downstream accuracy. The experimental results are shown in Figure 6 (Section 4.4).
>
> Furthermore, we would like to sincerely thank the reviewer for pointing out this promising direction. Motivated by the reviewer's suggestion, we conduct additional experiments under the same limited storage budget of 21.65 MB while varying the number of pruned teachers within this fixed budget. These new results provide a more comprehensive perspective on how the number of teachers influences downstream performance. The results are shown in the following table. $^*$ refers to the best single pruned teacher capacity with 7.2MB storage cost. The results are shown in the following table.
>
> |**# of Teachers**|Downstream Accuracy|$\sum_{\theta_{T_i}\in\Theta}\|\mathcal{C(\theta_{T_i})-C(\theta_{T}^*)}\|$|
> |-|-|-|
> |1 $^*$|47.5|-|
> |1|45.0|14.45MB|
> |2|48.6|7.15MB|
> |3|**48.8**|**~0MB**|
> |4|48.3|7.15MB|
> |5|47.2|14.35MB|
> |6|46.4|21.55MB|
> |7|45.2|28.75MB|
> |8|44.5|35.95MB|
> |9|43.6|43.15MB|
> |10|43.2|50.35MB|
>
>
> From the result, we observe a clear rise, then fall trend: increasing the number of pruned teachers initially improves accuracy, but further increasing the number eventually leads to performance degradation. This occurs because using too many teachers under a fixed budget forces each individual pruned teacher to operate with a very small portion of the available capacity, which significantly reduces its effectiveness and lowers the overall ensemble performance.
>
> In contrast, when the capacities of the pruned teachers are kept comparable and remain close to the capacity of the optimal single teacher, the ensemble is able to make full use of the available storage and achieves the best downstream performance. This observation aligns with the guideline we present in Eq.10.
>
>
> We sincerely appreciate the reviewer for bringing up this point, as it allows us to present the distinctions more clearly. We will revise the manuscript to ensure this is communicated unambiguously in the next version.
>
> ### [Questions 4] How is Acc in Algorithm 1 defined? Is it the accuracy of the model on the distilled dataset?
> We sincerely thank the reviewer for the problem. In Algorithm 1, Acc is the accuracy evaluated on the validation/test set, not on the distilled dataset.

---

> ### Author Response · Authors · 2025-11-29
> **Response to Reviewer fTh4 (part 3)**
>
> ### [Questions 5] The sensitive of the optimal teacher capacity to dataset domain or model architecture.
> We sincerely thank the reviewer for raising this important question. We appreciate the opportunity to investigate this aspect further and conducted additional experiments to examine the sensitivity across datasets and architectures. Following the reviewer's suggestion, we perform additional experiments on different architecture of the teacher model MobileNet-V2 and further evaluate the teacher capacity select on ImageNet-1K when transferring to ImageNet-100. Here, we adopt RDED as the base method and the experiments are conducted on the ImageNet-1K with IPC 1, 10, and 50 for architecture generalization, and ImageNet-100 for cross-dataset generalization. The experimental results are summarized below.
>
> Here, for cross architecture transferability, we also reuse exactly the same capacity ratio that the ResNet-18 architecutre is optimal on ImageNet-1K when experimenting on MobileNet-V2.
>
> |**MobileNet-V2**|**RDED**|**Ours**|
> |-|-|-|
> |IPC 1|2.7 $\pm$ 0.1|**12.5 $\pm$ 0.2**|
> |Extra Mem.|13.6MB|**0.25MB (~1.8%)**|
> |IPC 10|34.0 $\pm$ 0.2|**43.3 $\pm$ 0.3**|
> |Extra Mem.|13.6MB|**2.18MB (~16%)**|
> |IPC 50|53.3 $\pm$ 0.2|**54.9 $\pm$ 0.2**|
> |Extra Mem.|13.6MB|**3.25MB (~24%)**|
>
> Here, for cross dataset transferability, we also reuse exactly the same capacity ratio that is optimal on ImageNet-1K when experimenting on ImageNet-100. For example, under the IPC 1 setting, the optimal teacher capacity on ImageNet-1K corresponds to 1.8% of the base model, and we therefore adopt the same 1.8% capacity when conducting experiments on ImageNet-100.
>
> |**ImageNet-100**|**Ours (ori)**|**Ours (cross)**|
> |-|-|-|
> |IPC 1|10.9 $\pm$ 0.3 (0.1%)| 10.0 $\pm$ 0.2 (~1.8%)|
> |IPC 10|48.9 $\pm$ 0.3 (3.3%)|43.3 $\pm$ 0.3 (~16%)|
> |IPC 50|71.0 $\pm$ 0.2 (15.2%)|70.8 $\pm$ 0.1 (~24%)|
>
> *The numbers in parentheses denote the proportion of the teacher model's parameters with respect to the original model.*
>
> From the result, we compare the transferred capacity to the best capacity obtained by selecting directly on ImageNet-100. As shown in the above table, the performance gap is relatively small when the IPC is either very small (IPC 1) or relatively high (IPC 50), while a larger gap appears at IPC 10. Overall, this suggests that the teacher capacity obtained from ImageNet-1K has a reasonably broad robustness range.

---

### Official Review · Reviewer_Y37M · 2025-10-31

**Soundness:** 3
**Presentation:** 3
**Contribution:** 4
**Rating:** 6
**Confidence:** 5

**Summary:**

The paper introduces Flash-DD, a method to address a significant overhead in large-scale dataset distillation (DD). State-of-the-art (SOTA) DD methods often rely on soft labels from large, pre-trained "teacher models," which introduces substantial storage and computational costs, undermining the efficiency goals of DD. The authors' core contribution is the counter-intuitive finding that the full-sized, most accurate teacher model is suboptimal for guiding downstream training.

Flash-DD proposes a two-part solution:

1. DD-Oriented Parameter Reduction: A method to prune the teacher model not just to save space, but to find an "optimal capacity". This smaller teacher is shown to provide clearer class boundaries (Fig. 3) and leads to better downstream performance (Fig. 2). The method uses an iterative pruning process guided by a heuristic based on early-stage training loss.

2. Ensemble Label Generation: Guidelines for using any remaining storage budget to ensemble multiple, small, capacity-matched teachers, which further boosts performance by providing diverse perspectives.

Experiments show that Flash-DD can match SOTA performance using only 0.03% of the original teacher's storage while accelerating label generation by 843.81x. Alternatively, using a small 1.8% storage budget, it can improve SOTA accuracy by up to 13.4%. The method is shown to be a plug-and-play improvement for several existing DD methods (Table 1) and generalizes well to other student architectures and continual learning.

**Strengths:**

1. Critical Problem: Addresses a fundamental and practical flaw in SOTA large-scale DD methods: the massive storage and compute overhead of teacher models, which this paper correctly identifies as counter-productive.

2. Novel Core Insight: The paper's key contribution is the empirical discovery that downstream DD accuracy is a non-monotonic function of teacher capacity. The full-sized teacher is shown to be suboptimal, and a pruned, lower-capacity teacher provides superior guidance.

3. Exceptional Empirical Results: The efficiency gains are not marginal; they are orders of magnitude. Matching SOTA performance with 0.03% extra storage and an 843.81x speedup is a stellar result. Boosting SOTA accuracy by 13.4% with only 1.8% storage is equally powerful.

4. Generality: The method is demonstrated to be "plug-and-play," successfully improving the performance and efficiency of three different SOTA DD methods (Table 1).

5. Robustness: The authors provide strong evidence of generalization, showing their method works well when training diverse student architectures (including Transformers) and in a continual learning setting .

**Weaknesses:**

1. Weak Theoretical Justification: The paper's primary weakness. Proposition 1 is backed by a "proof" that is merely an intuition for a simplified linear case and holds no water for DNNs . This attempt at formalism is unconvincing and should be removed or heavily reframed as an "intuition."

2. Missing Ablation on Pruning Method: The method relies exclusively on $l_{1}$-norm magnitude pruning. This is a simple, unstructured method. It's an open question if the type of pruning matters. Would a more advanced, structured, or gradient-based pruning method find a better (or worse) teacher? The paper makes a tacit assumption that $l_1$ pruning is the right tool for this "knowledge-shaping" job, which is not ablated.

**Questions:**

1. Pruning Method Sensitivity: Have you investigated whether the choice of pruning method (e.g., $l_1$-norm vs. $l_2$-norm, or vs. a gradient-based method) impacts the "optimal capacity" or the final downstream performance? Is it possible that $l_1$ magnitude pruning is uniquely suited for this task?

2. Ensemble Saturation: The ensemble results (Fig. 6) are strong, showing 3 teachers are better than 2, which are better than 1 . Is there a point of diminishing returns? What happens if you use 5 or 10 teachers, constrained by the same total memory budget (meaning each teacher is even smaller)?

**Note to Authors: I am willing to raise my score if the authors can satisfactorily address the weaknesses and questions raised above in their rebuttal.**

---

> ### Author Response · Authors · 2025-11-29
> **Response to Reviewer Y37M**
>
> ### [Weaknesses 1] Proposition 1 should be removed or reframed as an "intuition."
> We sincerely appreciate the reviewer's thoughtful and constructive comment. Proposition 1 is intended to provide preliminary theoretical grounding for our formulation in a simplified linear setting. We will make this intended role clearer in the next version and articulate its scope more explicitly.
>
> ### [Weaknesses 2 & Questions 1] Ablation on Pruning Method, whether the choice of pruning method impacts the "optimal capacity" or the final downstream performance?
> We sincerely thank the reviewer for raising this excellent and insightful question. The possibility that different pruning criteria may influence the optimal teacher capacity or the downstream performance is indeed an important consideration.
>
> We would first like to clarify that the l1-norm pruning used in our main experiments corresponds to structured pruning. Following the reviewer’s suggestion, we conduct additional experiments to examine whether our framework is sensitive to the choice of pruning method. Specifically, we compared l1-norm pruning with l2-norm pruning and Taylor pruning (a gradient-based criterion). The results for ImageNet-1K across IPC settings 1, 10, and 50 are summarized in the following table.
>
> |**ImageNet-1K**|Taylor|l2-norm|l1 (in paper)|
> |-|-|-|-|
> |IPC 1|19.7 $\pm$ 0.2|19.6 $\pm$ 0.1|20.0 $\pm$ 0.1|
> |Extra Mem.|0.8MB|0.8MB|0.8MB|
> |IPC 10|47.3 $\pm$ 0.2|47.4 $\pm$ 0.1|47.5 $\pm$ 0.2|
> |Extra Mem.|7.2MB|7.2MB|7.2MB|
> |IPC 50|59.2 $\pm$ 0.1|59.1 $\pm$ 0.1|58.5 $\pm$ 0.1|
> |Extra Mem.|10.7MB|10.7MB|10.7MB|
>
> Across the different pruning criteria, all methods identify teacher capacities that are similar, and the corresponding downstream accuracies vary slightly. This consistency indicates that the capacity selection behavior and overall performance trends remain robust across pruning methods, suggesting that the framework is not strongly dependent on the specific pruning criterion.
>
>
> ### [Questions 2] Ensemble Saturation: The ensemble results (Fig. 6) are strong, showing 3 teachers are better than 2, which are better than 1. Is there a point of diminishing returns? What happens if you use 5 or 10 teachers, constrained by the same total memory budget (meaning each teacher is even smaller)?
> We sincerely thank the reviewer for this excellent question regarding the potential point of diminishing returns in the ensemble size. To investigate this more thoroughly, we conduct an extensive set of additional experiments on the ImageNet-1K IPC-10 setting, where we evaluate ensembles ranging from 1 to 10 pruned teachers under the same fixed memory budget 21.65 MB. $^*$ refers to the best single pruned teacher capacity with 7.2MB storage cost. The results are shown in the following table.
>
> |**# of Teachers**|Downstream Accuracy|$\sum_{\theta_{T_i}\in\Theta}\|\mathcal{C(\theta_{T_i})-C(\theta_{T}^*)}\|$|
> |-|-|-|
> |1 $^*$|47.5|-|
> |1|45.0|14.45MB|
> |2|48.6|7.15MB|
> |3|**48.8**|**~0MB**|
> |4|48.3|7.15MB|
> |5|47.2|14.35MB|
> |6|46.4|21.55MB|
> |7|45.2|28.75MB|
> |8|44.5|35.95MB|
> |9|43.6|43.15MB|
> |10|43.2|50.35MB|
>
>
> The results clearly show that the performance follows a rise-and-fall pattern. Increasing the number of teachers initially improves accuracy, but beyond a certain point the performance begins to decline. When the ensemble becomes too large, each individual pruned teacher receives only a very small portion of the fixed budget, resulting in significantly reduced predictive strength and consequently poorer ensemble performance. The best results are achieved when the capacities of all pruned teachers remain comparable and are close to the capacity of the optimal single teacher, which is consistent with the guideline provided in Equation (10).

---

### Official Review · Reviewer_94Eb · 2025-11-01

**Soundness:** 3
**Presentation:** 3
**Contribution:** 3
**Rating:** 4
**Confidence:** 3

**Summary:**

Flash-DD offers plug-and-play techniques for ultra-efficient soft label generation in dataset distillation (DD), countering teacher-model inefficiencies in SOTA large-scale methods (e.g., SRe2L, GVBSM, RDED). Innovations include DD-specific pruning to auto-determine teacher capacity (removing irrelevant params) and ensemble guidelines to repurpose extra space for stage-adapted knowledge diversity. This forms a storage-tunable paradigm: 0.03% extra storage yields 843.81× faster labeling at SOTA parity; 1.8% delivers up to 13.4% accuracy gains.

**Strengths:**

1. The findings in Figures 2 and 3 are interesting and inspiring.

2. The experimental results are promising.

**Weaknesses:**

Major Weakness:
1. The discussion in lines 247–251 regarding Figure 3 could be clarified. From the figure, it is somewhat difficult to discern which type of teacher model leads to better downstream performance.

2. The paper introduces several hyperparameters (e.g., $\lambda$), but the selection process and robustness analysis are not sufficiently discussed.

3. The motivation for Equation (10) is not clearly explained.

4. The paper lacks comparison with methods specifically designed for soft label compression (e.g., [1]), which would be important for a fair and comprehensive evaluation.

Minor Weakness:
1. Figure placement could be improved. For example, Figures 2 and 3 appear on page 3 but are first referenced on page 5. Similar inconsistencies occur elsewhere; a thorough check is recommended.

2. Figure and table captions are somewhat lengthy; using bold text to emphasize key points could improve readability.

3. Some captions contain minor inaccuracies—for instance, Figure 6 does not have “first row” and “second row.”

[1] Yu, Ruonan, et al. "Heavy labels out! dataset distillation with label space lightening." Proceedings of the IEEE/CVF International Conference on Computer Vision. 2025.

**Questions:**

1. Could the authors further elaborate on the observations and implications presented in Figure 3?

2. Could the authors provide more details on how the hyperparameters were chosen and whether the method is robust?

3. What is the underlying motivation behind Equation (10)?

4. Since label compression is an essential aspect of this work, could the authors include comparisons with label compression methods to strengthen the evaluation?

---

> ### Author Response · Authors · 2025-11-29
> **Response to Reviewer 94Eb**
>
> ### [Major Weakness 1 & Questions 1] Further elaborate on the observations and implications presented in Figure 3.
>
> We appreciate the reviewer's careful and professional evaluation. Figure 3 and Lines 247-251 aim to illustrate the relationships between the properties of teacher and downstream student models. To enhance clarity, we supplement the accuracies of downstream models trained by these four teachers as below:
>
> ||Downstream Acc|
> |-|-|
> |a|57.0|
> |b|58.5|
> |c|43.8|
> |d|13.4|
>
> We have refined the caption of the figure to improve readability. From these results:
>
> 1. The teacher must provide accurate and well-separated class predictions.
> When the teacher confuses classes, the student is unable to acquire reliable class-discriminative information, which directly limits downstream learning. As shown in Figure 3, only models (a) and (b) provide sufficiently clear guidance, whereas (c) and (d) introduce ambiguous or mixed signals that hinder effective training.
>
> 2. The teacher should produce informative soft labels that preserve both intra-class and inter-class structure.
> Although model (a) is highly accurate, its outputs are almost one-hot, resulting in overly sharp distributions that limit the student's ability to absorb relational cues among classes. In contrast, model (b) maintains high accuracy while offering softer probability distributions that encode meaningful cross-class structure.
>
> As stated in papers FRePo[1] and TESLA[2], hard one-hot labels restrict learning to a single target class, whereas soft labels preserve probability mass across multiple classes. These non-zero probabilities allow information to flow across classes and convey inter-class relationships that enhance the overall compression process. This idea is directly reflected in our observations: while the correct class should receive the highest probability, effective soft labels also retain small but non-zero probabilities for the remaining classes. These values capture the teacher's assessment of inter-class similarity and prevent non-target classes from collapsing to zero. Such distributions supply the student with additional relational information about how classes are positioned relative to one another, enabling a richer and more nuanced decision boundary than is possible with one-hot labels.
>
> Overall, an effective teacher model must strike a balance. **It should provide reliable class discrimination while producing sufficiently soft probability distributions.** The correct class should dominate, yet other classes should retain non-zero probabilities. This balance allows the student to learn both accurate class separation and meaningful inter-class relationships, ultimately leading to stronger downstream performance.
>
> [1] Dataset Distillation using Neural Feature Regression. Yongchao Zhou, Ehsan Nezhadarya, Jimmy Ba. Advances in Neural Information Processing Systems, 2022.
>
> [2] Scaling up dataset distillation to imagenet-1k with constant memory. Justin Cui, Ruochen Wang, Si Si, Cho-Jui Hsieh. International Conference on Machine Learning, 2023.

---

> ### Author Response · Authors · 2025-11-29
> **Response to Reviewer 94Eb (part 2)**
>
> ### [Major Weakness 2 & Questions 2] More details on hyperparameters chosen and the robustness of the method.
> We sincerely appreciate the reviewer's thoughtful comments and the opportunity to clarify the role of the hyperparameters in our method. Thank the reviewer for pointing out this important aspect of our design. Regarding $\lambda$, we would like to clarify that it appears only in the theoretical formulation, where it serves as a standard regularization coefficient to ensure the stability of the closed-form solution. It is not part of the trainable or tunable components of our method.
>
> For clarity, we would like to first clarify the definitions and roles of the key quantities used in our method.
>  1. $\tau(D_S)$ denotes a fixed constant value, as we define  and provide thereotical analysis in proposition 1: it is the downstream accuracy obtained when the student model is trained under the guidance of the original full-size teacher on the distilled dataset $D_S$, *i.e.*, the original baseline performance. This value is determined by the experimental setting without requiring any manual specification; it is not a tunable hyperparameter and does not require any additional training within our pipeline. For example, under the ImageNet-1K IPC-10 configuration, this value is 42.0 for RDED and 21.3 for SRe2L.
>  3. $\zeta$ indicates the selection range of teacher models, we define it in Eq. 8. It is naturally determined during the teacher parameter–reduction and selection process, based on the change in accumulated loss. Specifically, $\zeta$ corresponds to the "elbow" point of the accuracy-pruning rate curve for the teacher model, beyond which the accumulated loss increases only marginally. This quantity arises directly from the behavior of the accumulated-loss curve and therefore requires no manual specification or additional training.
>  2. $\gamma$ refers the contribution of the teacher model's own accuracy when determining the best teacher as stated in Eq. 7, $Dis(\theta_T)+\gamma Acc(\theta_T)$. We set $\gamma$ to make the accuracy term $Acc(\theta_T)$ and the accumulated-loss term $Dis(\theta_T)$ lie on comparable scales, ensuring that both factors contribute meaningfully when evaluating teacher capacity. Any reasonable value of $\gamma$ that brings the two terms onto the same scale is acceptable for this purpose. In paper, we set $\gamma=0.1$ for ImageNet-1K and Places365, and $\gamma=0.05$ for ImageNet-100 for all IPC settings. To further verify robustness, we include additional experiments showing that the method remains effective across a reasonable range of $\gamma$.
>
>  |Teacher Size (# Param)|$Dis(\theta_T)+\gamma Acc(\theta_T)$, $\gamma=0.05$|Downstream Acc|
>  |-|-|-|
>  |1.08|7.28|46.0 $\pm$ 0.1|
>  |0.50|7.22|46.4 $\pm$ 0.1|
>  |0.45|7.30|48.8 $\pm$ 0.2|
>  |0.37|**7.32**|**48.9 $\pm$ 0.3**|
>  |0.33|6.99|48.2 $\pm$ 0.2|
>  |0.26|6.56|48.7 $\pm$ 0.2|
>  |0.22|6.49|48.2 $\pm$ 0.1|
>
> *The experiment is conducted on ImageNet-100 IPC 10, $\gamma=0.05$. Here, teacher with the highest $Dis(\theta_T)+\gamma Acc(\theta_T)$ value is considered as the optimal capacity model for downstream tasks.*
>
>  |Teacher Size (# Param)|$Dis(\theta_T)+\gamma Acc(\theta_T)$, $\gamma=0.04$|Downstream Acc|
>  |-|-|-|
>  |1.08|6.51|46.0 $\pm$ 0.1|
>  |0.50|6.51|46.4 $\pm$ 0.1|
>  |0.45|6.61|48.8 $\pm$ 0.2|
>  |0.37|**6.65**|**48.9 $\pm$ 0.3**|
>  |0.33|6.33|48.2 $\pm$ 0.2|
>  |0.26|5.93|48.7 $\pm$ 0.2|
>  |0.22|5.88|48.2 $\pm$ 0.1|
>
>  *ImageNet-100 IPC 10, $\gamma=0.04$.*
>
>  |Teacher Size (# Param)|$Dis(\theta_T)+\gamma Acc(\theta_T)$, $\gamma=0.03$|Downstream Acc|
>  |-|-|-|
>  |1.08|5.75|46.0 $\pm$ 0.1|
>  |0.50|5.80|46.4 $\pm$ 0.1|
>  |0.45|5.91|48.8 $\pm$ 0.2|
>  |0.37|**5.97**|**48.9 $\pm$ 0.3**|
>  |0.33|5.67|48.2 $\pm$ 0.2|
>  |0.26|5.30|48.7 $\pm$ 0.2|
>  |0.22|5.27|48.2 $\pm$ 0.1|
>
>  *ImageNet-100 IPC 10, $\gamma=0.03$.*
>
>  |Teacher Size (# Param)|$Dis(\theta_T)+\gamma Acc(\theta_T)$, $\gamma=0.02$|Downstream Acc|
>  |-|-|-|
>  |1.08|4.99|46.0 $\pm$ 0.1|
>  |0.50|5.10|46.4 $\pm$ 0.1|
>  |0.45|5.22|48.8 $\pm$ 0.2|
>  |0.37|**5.30**|**48.9 $\pm$ 0.3**|
>  |0.33|5.01|48.2 $\pm$ 0.2|
>  |0.26|4.67|48.7 $\pm$ 0.2|
>  |0.22|4.67|48.2 $\pm$ 0.1|
>
>  *ImageNet-100 IPC 10, $\gamma=0.02$.*
>
>
>  4. $\mu$, it controls the contribution of each pruned teacher to form the weighted ensemble output as shown in Eq. 9. During ensembling, each pruned teacher produces a full prediction independently, and the final supervision signal is obtained by taking a weighted average of these predictions using the coefficients $\mu_i$. The $\mu_i$ are initialized uniformly as $\mu_i=\frac{1}{|\Theta|}$, $|\Theta|$ is the number of pruned teachers contributed to the emsembled output. Here, following the guidelines stated in Eq. 10, since the pruned teachers are turned to have similar sizes, and this setting already provides stable performance.
>
>
> We are truly grateful to the reviewer for these insightful questions. They greatly help us refine the presentation and provide a clearer and more complete analysis.

---

> ### Author Response · Authors · 2025-11-29
> **Response to Reviewer 94Eb (part 3)**
>
> ### [Major Weakness 3 & Questions 3] The underlying motivation behind Equation (10).
> We sincerely thank the reviewer for the thoughtful and constructive question, as well as for the careful reading of our manuscript. We truly appreciate the opportunity to further clarify the motivation behind Equation (10).
>
> The core motivation of Equation (10) is to more effectively utilize the available space when the storage budget becomes larger, **as we state in lines 320-323, 359-360**. Our proposed method fully utilizes different available storage budgets in a parameter-efficient manner, and it contains two complementary stages. Under highly constrained storage budgets, such as the ImageNet-1K IPC-10 setting, our method first identifies a single pruned teacher with an appropriate capacity that provides strong downstream performance. As the available budget increases, however, keeping the same teacher capacity leaves a substantial portion of the space unused, which limits the potential performance improvement. To make better use of the additional capacity, we introduce Equation (10) as a principled guideline for reorganizing the parameter budget across multiple pruned teachers. This strategy enables the expanded budget to be allocated more effectively and leads to improved supervision quality in the downstream task.
> More specifically, Equation (10) incorporates several constraints that together maintain the effectiveness and stability of the ensemble of sub-teachers.
> 1. The first two constraints limit how much the capacities of sub-teachers can differ from one another. This prevents their predictions on the same sample from diverging excessively, thereby avoiding inconsistent or noisy guidance.
> 2. The third constraint requires each sub-teacher to maintain an accuracy above the threshold defined earlier, which helps prevent incorrect supervision and preserves the overall quality of the ensemble.
> 3. The fourth constraint encourages each sub-teacher's capacity to remain close to that of the optimal single teacher, ensuring that the additional parameter space is used effectively rather than being negatively affected by an excessive number of low-capacity models.
>
> The effectiveness of this formulation, and the role of these constraints, is further supported by our experiments, as shown in Figure 6. In addition, we conduct another set of experiments under the ImageNet-1K IPC-10 setting, where we varied the ensemble size from 1 to 10 teachers while keeping the total storage budget as 21.65MB, $^*$ refers to the best single pruned teacher capacity with 7.2MB storage cost. The results are shown in the following table.
>
> |**# of Teachers**|Downstream Accuracy|$\sum_{\theta_{T_i}\in\Theta}\|\mathcal{C(\theta_{T_i})-C(\theta_{T}^*)}\|$|
> |-|-|-|
> |1 $^*$|47.5|-|
> |1|45.0|14.45MB|
> |2|48.6|7.15MB|
> |3|**48.8**|**~0MB**|
> |4|48.3|7.15MB|
> |5|47.2|14.35MB|
> |6|46.4|21.55MB|
> |7|45.2|28.75MB|
> |8|44.5|35.95MB|
> |9|43.6|43.15MB|
> |10|43.2|50.35MB|
>
>
> The additional experiment on ImageNet-1K IPC-10 further confirms the motivation behind Equation (10). As we vary the ensemble size from one to ten teachers under the same storage budget, the accuracy first increases and then decreases. When the ensemble becomes too large, each pruned teacher receives insufficient capacity, which reduces its effectiveness. The best results consistently occur when all pruned teachers have capacities that are close to one another and close to the optimal single-teacher capacity, exactly reflecting the capacity-balancing principle described in Equation (10).

---

> ### Author Response · Authors · 2025-11-29
> **Response to Reviewer 94Eb (part 4)**
>
> ### [Major Weakness 4 & Questions 4] Comparison with label compression methods to strengthen the evaluation.
> We sincerely appreciate the reviewer's insightful suggestion. In response, we conduct comparisons with HeLlO [1] on ImageNet-1K with IPC 1, 10, and 50. We additionally evaluate cross-architecture generalization performance using MobileNet-V2 and EfficientNet-B0. The corresponding results are provided in the table below. For fairness, we set the same storage budget specified in HeLlO, and our results under this setting are denoted as ours $^*$ in the table.
>
> |**ImageNet-1K**|HeLlO [1]|**Ours $^*$**|**Ours**|
> |-|-|-|-|
> |IPC 1|12.9 $\pm$ 0.3|**21.5 $\pm$ 0.1**|**20.0 $\pm$ 0.1**|
> |Extra Mem.|3.3MB|**3.3MB**|**0.8MB**|
> |IPC 10|43.7 $\pm$ 0.1|**45.5 $\pm$ 0.1**|**47.5 $\pm$ 0.2**|
> |Extra Mem.|3.3MB|**3.3MB**|**7.2MB**|
> |IPC 50|52.2 $\pm$ 0.1|**52.5 $\pm$ 0.1**|**58.5 ±0.1**|
> |Extra Mem.|3.3MB|**3.3MB**|**10.7MB**|
>
>
>
> Cross architecture performance:
> |**ImageNet-1K**|**Extra Mem.**|**MobileNet-V2**|**EfficientNet-B0**|
> |-|-|-|-|
> |HeLlO|3.3MB|26.5 $\pm$ 0.2|38.1 $\pm$ 0.5|44.4 $\pm$ 0.2|
> |**Ours $^*$**|**3.3MB**| **41.2 $\pm$ 0.2** |**45.2 $\pm$ 0.1**|
> |**Ours**|**7.2MB**|**41.5 $\pm$ 0.5**|**47.4 $\pm$ 0.1**|
>
> Across all settings, our proposed method achieves consistently strong performance under the same storage budget, and additional gains are observed when more budget is available. We sincerely thank the reviewer again for recommending this valuable comparison.
>
> ### [Minor Weaknesses 1 & 2 & 3] Improvement of figure placement and caption.
> We thank the reviewer for the careful and attentive review. We will revise the paper accordingly in the next version based on these professional suggestions.

---

### Author Response · Authors · 2025-12-02
**Summary for Reviews and Rebuttals**

Dear Reviewers, AC, SAC, and PC,

We sincerely appreciate the time, effort, and thoughtful consideration that you and all reviewers devoted to evaluating our submission. We are grateful for the constructive feedback and for the recognition of the importance of our research direction (**Reviewers Y37M, fTh4**), the novelty of our core insights (Reviewer Y37M), the effectiveness of our method (**Reviewers 94Eb, Y37M, fTh4, z3eu**), the promise of our empirical results (**Reviewers 94Eb, Y37M, fTh4, z3eu**), the interesting and inspiring findings (**Reviewer 94Eb**), the generality of our method (**Reviewer Y37M**), its robustness (**Reviewer Y37M**), and the thoroughness of our experimental evaluation (**Reviewers Y37M, fTh4**).

**Reviewers Y37M, fTh4, and z3eu** provided positive scores, and both **Reviewers Y37M and fTh4** explicitly indicated their willingness to raise their scores (*"I am willing to raise my score"* and *"I like the paper overall and would be happy to raise my score"*). We submitted detailed rebuttals and additional experiments addressing all concerns, but unfortunately the discussion system closed before they were able to participate in the discussion phase.

**Reviewer 94Eb** initially assigned a score of 4, raising concerns about: (1) the interpretation of observations in Figure 3; (2) hyperparameter settings; (3) the underlying motivation of Eq. 10; and (4) additional comparisons with the baseline HeLlO.
- To address (1), we provided comprehensive explanations and revised the caption of Figure 3 (in red) to enhance clarity and readability.
- To address (2), we clarified that $\gamma$ is the only hyperparameter requiring manual selection and added experiments with different $\gamma$ values to demonstrate the robustness of our method.
- To address (3), we offered a detailed explanation of the motivation behind Eq. 10, consistent with the discussion in our original submission (lines 320–323, 359–360).
- To address (4), we included additional comparisons with HeLlO, showing that our approach consistently outperforms this baseline across diverse experimental settings.

We sincerely appreciate the reviewers' careful consideration throughout the process, and we genuinely hope that the clarifications provided here will be helpful in the final assement.

Best regards,

Authors of submission 8479

---

### Meta-Review · Area_Chair_35pK · 2025-12-21

**Summary:**

* One critical aspect missed by reviewers is the mechanism of soft labeling in dataset distillation. The soft labels are not generated during training, but before training and after the distilled data is acquired. Therefore, the major contribution of this paper is much less significant. The method still leads to the same amount of soft labels for downstream training, without actual resource saving.
* The t-SNE distribution in Figure 3 is hard to interpret. No explicit rules can be concluded from the figures for which model performs the best for downstream tasks.
* Based on the pseudo code in Algorithm 1, $\tau(D_S)$ is required before pruning. $\tau(D_S)$ is the accuracy of the downstream student models trained with the guidance of the original pre-trained full-size teacher model on the distilled dataset $D_S$. Based on the first concern, where soft labels are generated offline, the proposed method actually increases the required computation for soft label generation. It conflicts with the efficiency claim.
* The theoretical analysis is heuristic. The simplification of the linear network is trivial for this formulation, where the actual neural networks are mostly non-linear. Many steps are wrong. For example, the context is a classification problem, but the test error in Equation 4 is for regression. The conclusion of $\varepsilon(S_1)=\varepsilon(S_0)$ doesn't make sense for the proposition, either. Does that mean the test error is not influenced by $T_1$?
* The paper is poorly presented. The citation format and caption format are wrong. Many figures use font sizes that are too small. There is no explanation or interpretation for Figure 4. There is no detail provided for how the model is pruned. Terms like "ACC" and "self-ACC" are hard to understand when used in different contexts. I see a major revision required for improving and clarifying these necessary details.
* Some ablation studies and extra comparisons were requested by reviewers, including the pruning method, ensemble saturation, hyperparameter analysis, and label compression methods.
* The sensitivity of the optimal teacher capacity to the dataset domain or model architecture.

**Reviewer Concerns:**

* The authors provided a comparison with the label compression method HeLlO. However, the proposed method uses the full soft label set without compression. The comparison is not exactly fair. A fairer demonstration is applying the label compression method on top of the proposed method to validate the robustness of the generated soft labels.
* The authors provided some explanation on the t-SNE interpretation. However, it is still very vague and heuristic, without careful examination of underlying mechanisms.
* Hyperparameter analysis is provided, but only within a very small range. Hyperparameter analysis is not to expose the weakness of the method, but to help better understand the mechanism.
* The theoretical analysis is still heuristic and based on empirical observations, which was raised as the primary concern of reviewer Y37M.
* The authors provided experimental results to demonstrate that the pruning ratio acquired with one setting can be generalized to some other settings. It then raises another question: Why, in Table 1, do different settings have different pruning ratios? For the same IPC, ImageNet-1K demands a significantly larger ratio than the other two datasets. How much difference does scaling the ratio from 3.3% to 16.1%?

**Reviewer Scores:**

This paper provides an interesting perspective that the model architecture needs to be carefully designed for soft label generation. However, the insights are not grounded or explained well in the current version. Some major concerns raised by the reviewers, such as the theoretical analysis, t-SNE observation, and additional empirical results, are not fully addressed in the rebuttal. I do not think the reviewers would raise scores based on the current version of the rebuttal and revision.

While the paper originally received overall positive ratings, I would have to recommend rejection before a major revision that further clarifies vague points in the paper and provides clearer insights for subsequent explorations.

---

### Decision · Program_Chairs · 2026-01-26

Reject